# Assessment of Individual Radiosensitivity in Breast Cancer Patients Using a Combination of Biomolecular Markers

**DOI:** 10.3390/biomedicines11041122

**Published:** 2023-04-07

**Authors:** Matus Durdik, Eva Markova, Pavol Kosik, Katarina Vigasova, Sachin Gulati, Lukas Jakl, Katarina Vrobelova, Marta Fekete, Ingrid Zavacka, Margita Pobijakova, Zuzana Dolinska, Igor Belyaev

**Affiliations:** 1Department of Radiobiology, Cancer Research Institute, Biomedical Research Center v.v.i., Slovak Academy of Sciences, 845 05 Bratislava, Slovakia; 2Department of Radiation Oncology, Radiological Centrum, National Cancer Institute, 812 50 Bratislava, Slovakia

**Keywords:** breast cancer, DNA repair γH2AX/53BP1 foci, apoptosis, Metafer, human lymphocytes, radiotherapy, chemotherapy, ionizing radiation

## Abstract

About 5% of patients undergoing radiotherapy (RT) develop RT-related side effects. To assess individual radiosensitivity, we collected peripheral blood from breast cancer patients before, during and after the RT, and γH2AX/53BP1 foci, apoptosis, chromosomal aberrations (CAs) and micronuclei (MN) were analyzed and correlated with the healthy tissue side effects assessed by the RTOG/EORTC criteria. The results showed a significantly higher level of γH2AX/53BP1 foci before the RT in radiosensitive (RS) patients in comparison to normal responding patients (NOR). Analysis of apoptosis did not reveal any correlation with side effects. CA and MN assays displayed an increase in genomic instability during and after RT and a higher frequency of MN in the lymphocytes of RS patients. We also studied time kinetics of γH2AX/53BP1 foci and apoptosis after in vitro irradiation of lymphocytes. Higher levels of primary 53BP1 and co-localizing γH2AX/53BP1 foci were detected in cells from RS patients as compared to NOR patients, while no difference in the residual foci or apoptotic response was found. The data suggested impaired DNA damage response in cells from RS patients. We suggest γH2AX/53BP1 foci and MN as potential biomarkers of individual radiosensitivity, but they need to be evaluated with a larger cohort of patients for clinics.

## 1. Introduction

Approximately half of all cancer patients undergo radiotherapy (RT), alone or in combination with chemotherapy (CHT), which may result in severe acute/late side effects in ~5% of patients [1]. Acute side effects in breast cancer patients mainly include skin reactions, such as erythema, redness and desquamations [2,3]. Late side effects, such as hyperpigmentation, fibrosis [4] and even a higher risk of pulmonary fibrosis, can also present later after RT [5]. Thus, it is of great importance for clinics to identify these radiosensitive (RS) patients at an increased risk for side effects. For now, no effective method has been introduced to clinical practice to identify RS patients with clinically sufficient sensitivity and specificity. The most attention has been given to quantification of DNA double strand breaks (DSBs) using DNA repair foci [2,3,6,7,8], which are formed by proteins, such as phosphorylated histone H2AX (γH2AX) [9] and the tumor suppressor p53 binding protein 1 (53BP1) [10]. Some residual foci can remain in the cells for a relatively long time and may correlate with individual radiosensitivity [11]. Radiation-induced apoptosis was also investigated as a possible biomarker for assessing radiosensitivity in vitro prior to RT [4,12,13,14,15,16]. In particular, our previously published results demonstrated negative correlation between the accumulation of late apoptotic/necrotic (LAN) cells and the level of 53BP1 and positive correlation with individual RS [2]. Micronuclei (MN) and chromosomal aberrations (CAs) represent other markers with possible clinical usability as far as a number of studies have shown differences between RS patients and normal responders (NOR) after in vitro irradiation of their lymphocytes using the CA assay [17,18] and also cytokinesis-block micronucleus (CBMN) assay [19,20,21]. In the present study, we analyzed the individual radiosensitivity of two groups of BC patients with either combined CHT/RT or RT alone treatment in relation to DNA repair foci, apoptosis, CA and MN in their lymphocytes. Analysis of these two groups was performed in order to assess the effects of the CHT treatment on genomic instability and apoptosis induced by RT. We assessed DNA repair foci, apoptosis, CA and MN before, during and after the RT, with the aim to compare the level of residual DNA damage and genomic instability in the RS and NOR patients. Furthermore, we analyzed apoptosis in specific populations of blood mononuclear cells (MNCs), namely neutrophils and monocytes, which have important roles in managing inflammation and phagocytosis, as well as in lymphocytes and also hematopoietic stem/progenitor cells that serve as a basis for renewing the pool of blood cells. To further validate our data, we performed a set of in vitro experiments by analyzing the kinetics of radiation-induced γH2AX/53BP1 foci and apoptosis.

## 2. Materials and Methods

### 2.1. Ethical Considerations

The study was approved by the Institutional Review Board of the National Cancer Institute (NCI), Bratislava. Written informed consent was obtained from each participant prior to collection of peripheral blood samples. The Ethics Committee approved this consent procedure.

### 2.2. Chemicals

Reagent-grade chemicals were obtained from Merck-Sigma (Burlington, MA, USA) and Thermo Fisher Scientific (Waltham, MA, USA).

### 2.3. Patients and Cells

Fifty-seven BC patients (mean age 59.2 ± 11.4) were enrolled in this study after signing the informed consent (Appendix A). Twenty-six patients (mean age 61.1 ± 9.2 year) were treated with RT alone by using linear accelerator Elekta Versa and Elekta Synergy Platform (Elekta, Stockholm, Sweden). The remaining 31 out of these 57 patients (mean age 57.4 ± 12.8 years) were subjected to CHT with various chemotherapeutics, such as Paclitaxel, Docetaxel, Herceptin, Doxorubicin or FEC (combination 5-Fluorouracil, Epirubicin and Cyclophosphamide), before RT. CHT treatment was usually finished one month before the start of RT. Total dose per patient varied from 40.5 up to 50 Gy (mean 44.5 ± 3.5 Gy), with the mean dose per fraction 2.37 ± 0.23 Gy. One patient (no. 49) acquired only 30 Gy (12 fractions) because RT had to be discontinued due to severe acute toxicity. The number of irradiation fractions ranged from 15 to 25, with a mean number of 19.2 fractions per patient. The patients were irradiated in a 4- or 5-week cycle. 

### 2.4. In Vitro Irradiation

We collected peripheral blood samples from the BC patients two days before the start of RT, during RT (24 h after the 4th fraction), one month and one year after the end of RT. These time points were shown to be the most appropriate for analysis of kinetics of genomic instability before, during and after the RT based on our previous study [2]. The dose acquired by blood cells during irradiation is not homogenous because of blood circulation, and the time of reaching equilibrium between irradiated and non-irradiated cells is individual. Therefore, we chose to draw the blood the next morning after irradiation, when relative equilibrium in blood and plateau in kinetics of residual foci are usually observed. MNCs were isolated from fresh blood via centrifugation on a lymphocyte separation medium (LSM, Sigma Aldrich, Burlington, MA, USA) as previously described [2]. Adherent monocytes were removed from the samples via the incubation of MNC for 2 h in RPMI medium supplemented with 10% FBS and 100 IU/mL penicillin, 100 μg/mL streptomycin, at 37 °C in a 5% CO_2_ incubator. Viability of the remaining lymphocytes was at least 95%, as defined by the Trypan blue exclusion assay. The acute adverse reactions of all patients such as acute skin erythema reactions and other markers of adverse side effects were classified according to the Radiation Therapy Oncology Group (RTOG)/European Organization for Research and Treatment of Cancer (EORTC) criteria side effects [22]. Twelve patients exhibited no adverse reactions (G0); thirty-five patients had G1 reactions (follicular, erythema). Five G2 patients exhibited severe acute side effects (erythema, skin desquamations) as assessed during the study. The G0 - 1 patients were classified as normal responders (NOR) and the G2 patients as RS. Late healthy tissue side effects, such as hyperpigmentation (n = 26), erythema (n = 3) and telangiectasia (n = 2), edema and swelling (both n = 1), were evaluated 1 year after the therapy using the RTOG criteria and data from the experimentally defined end-points were compared in the patients with or without late side effects (Appendix A). Telangiectasia, edema and swelling were not accompanied in the statistical analysis since there were too few cases.

### 2.5. In Vitro Irradiation

For in vitro experiments, we used frozen peripheral blood lymphocytes (PBLs) collected before the RT from four RS and four NOR breast cancer patients. The patients were matched according to age and CHT status. Two RS and two NOR patients were treated with CHT and RT while another two RS and NOR were treated with RT alone. The cells were thawed and incubated as previously described [23]. Samples were irradiated with 2 Gy of γ-rays using TruBeam linear accelerator (Varian medical systems, Palo Alto, CA, USA) at the St. Elizabeth hospital, Bratislava. We analyzed γH2AX/53BP1 DNA repair foci 30 min, 2 h and 24 h after irradiation. Radiation-induced and endogenous apoptosis was analyzed 0, 24 and 48 h post-irradiation.

### 2.6. γH2AX/53BP1 DNA Repair Foci

Microscopic slides for DNA repair foci analysis were prepared as previously described [23]. Briefly, two hundred thousand cells were cytospan onto each of the two fields of the polysine-coated microscopic slide (ThermoShandon, Thermo Fisher Scientific, Runcorn, UK). Afterwards, cells were fixed in 3% paraformaldehyde for 15 min and stored in PBS until immunostained. For this, the slides were incubated with 0.1% Triton X-100, washed with PBS and then the cells in each field were blocked by dropping 100 μL of 5% bovine serum albumin and incubating for 30 min. Next, the cells were immunostained with anti-γH2AX (mouse monoclonal) and anti-53BP1 (rabbit polyclonal) antibodies (Novus biologicals, Cambridge, UK), in concentrations of 1:400 and 1:1000, respectively. Secondary antibodies, Alexa Fluor 488 anti-mouse and 555 anti-rabbit in a 1:250 concentration were used for visualizing DNA repair foci. Finally, the slides were mounted with Vectashield medium containing DAPI (4′,6-diamidino-2-phenylindole) (Vector laboratories, Burlingame, CA, USA), sealed with nail polish and stored in fridge until analysis. The slides were scanned using a Metafer Slide Scanning System Version 3.6 (MetaSystems, Altlussheim, Germany) with a Zeiss Axio Imager Z1 microscope (Zeiss Microscopy, Jena, Germany) using 63× objective. Cell images were acquired automatically by the custom-made classifier in seven focal planes with 1 µm distance from each other. γH2AX, 53BP1 and co-localizing γH2AX/53BP1 foci were enumerated in at least 150 cells per one slide through visual inspection of cell galleries using the Metafer 5 software.

### 2.7. Apoptosis

Late apoptotic/necrotic (LAN) cells were analyzed in blood samples from BC patients before, during and after RT using TruCount tubes (BD biosciences, San Jose, CA, USA) according to the manufacturer’s protocol. Briefly, CD34-PE/CD45-FITC antibody mix and 7-AAD (BD biosciences) were added to 100 µL of whole blood in dilutions of 1:10 and 1:25, respectively. Before the analysis, erythrocytes were lysed using 1 mL of the 10% BD Lysis buffer. Samples were analyzed using the FACS Canto II flow cytometer (BD biosciences) immediately after lysis. Lymphocytes, monocytes and neutrophils were gated according to different CD45 expression profiles, and HSPCs were gated based on the CD34 expression. Percentage of LAN cells in each population was assessed using 7-amino actinomycin D (7-AAD). 

In in vitro experiments, radiation-induced and endogenous apoptosis was analyzed using Annexin-V/7-AAD assay as previously described [24]. Briefly, 2 × 10^5^ cells were taken from each sample and, after washing with PBS, incubated for 30 min with the Annexin-V-FITC (1:40), 7-AAD (1:50) and CD45-Pacific blue (1:40) in 100 μL of the Annexin binding buffer (Merck-Sigma). Then, cells were washed again with PBS, resuspended in 200 μL of the Annexin buffer and analyzed using the BD FACS Canto II flow cytometer. At least 20,000 cells were acquired per each treatment condition. Compensations were performed based on single antibody stain controls. 

### 2.8. Chromosomal Aberrations and Cytokinesis-Blocked Micronuclei (CBMN) Assay

Chromosomal aberrations and micronuclei were analyzed after activation of T-cell division with phytohemagglutinin (PHA, Merck-Sigma). In the CBMN assay, cytochalasin B (Merck-Sigma) was added 44 h after PHA activation. Colchicin (Merck-Sigma) was added 71 h and 15 min (45 min before fixation) after the induction of cell division with PHA for CA analysis. Harvesting, hypotonization and fixation were performed for the CA and CBMN assays as previously described [25,26]. In the CA assay, metaphases were detected automatically using Metafer software (MetaSystems), and then 1000 metaphases per sample were analyzed manually for dicentrics and ring chromosomes. In the CBMN assay, binucleated cells were detected automatically using Metafer software, and then micronuclei were enumerated in 1000 binucleated cells.

### 2.9. Statistical Analysis

Analysis of variance (ANOVA) and comparison between treatment conditions (Fisher LSD, *t*-test) were carried out using Statistica 8.0 (Statsoft, Dell, Round Rock, TX, USA). The results were considered significantly different at *p* < 0.05. Receiver operating characteristic (ROC) analysis was carried out using MedCalc software (MedCalc software Ltd., Ostend, Belgium). Area under the ROC curve was used to determine specificity and sensitivity of assessing individual radiosensitivity. Calculations of statistical power were performed using online calculator Clincalc: www.clincalc.com (accessed on 20 March 2023).

## 3. Results

### 3.1. DNA Repair Foci in PBL of BC Patients

We analyzed γH2AX/53BP1 foci in the PBL of fifty-seven BC patients at the beginning of RT, 24 h after the 4th fraction of RT and one month as well as one year after the RT end. Of the 57 patients, 5 were categorized as RS with grade 2 erythema and patchy desquamations (G2 acute reactions according to the RTOG/EORTC criteria). From these five RS patients, CHT treatment was applied to three RS patients prior to RT while another two RS patients were not treated with CHT. Thus, the prevalence of RS was almost the same in the groups of patients treated with CHT+RT (31 patients) and RT alone (26 patients). First, we performed the multifactorial ANOVA analysis of the complete data pool collected at the beginning of RT, 24 h after the 4th fraction of RT and one month after the RT. We found a significant effect from acute individual radiosensitivity, sample collection time and CHT (*p* = 0.001, 0.000004 and 0.006, respectively). Further analysis of the pooled data from all time points via the multiple comparison Fisher LSD test showed significantly increased levels of 53BP1 and a trend towards an elevated level of γH2AX foci before RT in the RS patients compared to the NOR group (*p* = 0.002 and 0.09 for 53BP1 and γH2AX, respectively) (Figure 1 and Appendix A). No difference between groups was observed for γH2AX/53BP1 foci, either due to low co-localization (approximately 30%) or because the most residual γH2AX and residual 53BP1 correspond to different types of damage. At the later timepoints, namely 24 h after the 4th RT fraction and one month and one year after the course of RT, we did not find any statistically significant difference in the level of DNA repair foci between RS and NOR patients (Figure 1). 

Using the ROC analysis, we checked whether the levels of endogenous γH2AX and 53BP1 foci before the start of RT were sensitive and specific enough for identification of the radiosensitive patients. Area under the curve (AUC) was 0.77 and 0.75 for 53BP1 and γH2AX, respectively, suggesting that analysis of either γH2AX or 53BP1 is only moderately sensitive/specific and not sufficient enough to identify radiosensitive patients prior to RT in clinical settings (Appendix A). 

In line with previously published data [2,3], we found significant induction of co-localizing γH2AX/53BP1 foci 24 h after the 4th fraction (Fisher LSD, *p* = 0.025) based on the analyzed pooled data from all patients and all timepoints (Figure 1). This induced level then significantly decreased one month and one year after RT (Fisher LSD, p = 0.002 and 0.014, respectively). Although the trend to induction of γH2AX and 53BP1 foci was observed by 24 h after the 4th fraction, it was statistically insignificant. The numbers of γH2AX or 53BP1 foci decreased after the completion of RT (Fisher LSD, *p* ≤ 0.03). In the NOR group, we obtained the same results as in the group of all patients, since 52 out of 57 patients belonged to the NOR group (Figure 1). Interestingly, in the group of RS patients, we found a significantly higher amount of 53BP1 foci before the start of RT in comparison to any other collection timepoint: 24 h after the 4th fraction, one month and one year after the RT (Fisher LSD, *p* = 0.05, 0.02 and 0.002, respectively). No such trend was observed in the NOR group. Individual variability did not change, both during and after the RT, as illustrated by the 95% confidence interval in Figure 1.

Further, we then compared the effects in cells of patients treated with CHT and RT and from those patients treated with RT alone. Although in the dataset pooled from all patients, we found significant induction of γH2AX/53BP1 foci 24 h after the 4th fraction of RT, only a trend to this induction was seen in datasets from CHT + RT and RT alone patients. This difference was likely caused by higher statistical power for the dataset pooled from all patients. Next, we found a higher level of co-localizing γH2AX/53BP1 foci before the RT in the CHT + RT group as compared with the RT alone group (Fisher LSD, *p* = 0.04). These data suggested persistence of CHT-induced damage until the start of RT. In the group of CHT + RT patients, results also showed a significant decrease in γH2AX foci detected one year after the RT in comparison to the numbers counted before the RT (Fisher LSD, *p* = 0.03). There was also a trend to a decreased level of co-localizing γH2AX/53BP1 foci one month and year after RT, as compared to values observed before the start of RT (Fisher LSD, *p* = 0.089 and 0.057, respectively). However, no such trend was present in the RT only group, probably mirroring the CHT-induced foci that further decreased. Since there was no CHT, in the RT alone group, we did not see a decrease in foci count after the RT (Figure 2). To conclude, the data suggested persistence of CHT-induced DNA damage in the PBL until one month after the CHT treatment. We also concluded that CHT-induced DNA damage persisted in the lymphocytes of BC patients at least one month after CHT and that CHT treatment is more genotoxic to PBL in comparison to RT.

We also analyzed DNA repair foci in patients with late toxicity, specifically hyperpigmentation (n = 26) and erythema (n = 3), in comparison to patients without any symptoms of late toxicity. The number of foci was the same in both groups (Appendix A). In conclusion, we did not observe any difference in the level of γH2AX, 53BP1 and co-localizing γH2AX/53BP1 foci between the patients with or without late toxicity. 

### 3.2. Cell Death in MNC Populations of BC Patients

We studied possible correlation between radiosensitivity of BC patients and percentage of LAN cells (7-AAD +) in different populations of MNC, namely: lymphocytes, monocytes, neutrophils and HSPC (Figure 3).

Statistical analysis of results by the multivariate and univariate ANOVA did not show any differences in the amount of LAN cells between RS and NOR patients before (Figure 4), during and after the course of RT. No correlation was found between the percentage of LAN cells and either type of foci. Analysis of data from patients treated solely with RT and those treated with CHT + RT using multivariate and univariate ANOVA did not reveal any significant difference between these two groups of patients in either of the studied populations at any time of blood collection. Interestingly, we observed a significantly lower level of viable cells in neutrophils in comparison to lymphocytes and monocytes and also in HSPC as compared to lymphocytes and monocytes (Fisher LSD test *p* < 0.001, at all tested conditions). These data indicate that neutrophils and HSPCs are more sensitive to RT treatment.

However, there was no increase in the level of LAN cells during and after the RT in any of studied populations, suggesting the removal of apoptotic cells through phagocytosis. We conclude that there is no difference in the level of LAN cells between RS and NOR patients and analysis of the in vivo-defined LAN cells in either MNC or any selected cell population is not a sensitive marker for the assessment of individual radiosensitivity. LAN cells did not correlate with late healthy tissue toxicity either. 

### 3.3. MN and Chromosomal Aberrations in PBL of BC Patients

We also analyzed micronuclei and CA (dicentric and ring chromosomes) in PBL of thirteen BC patients subjected to RT (Appendix A). Six BC patients were treated with CHT prior to RT while seven patients were not. Of these 13 patients, 2 developed G2-stage adverse skin reactions after RT. Neither of these RS patients were treated with CHT. We observed a trend to higher MN frequency in RS patients before the RT, 2.2 vs. 1.4 in NOR patients. However, this trend was not statistically significant. Additionally, the frequency of total CA, rings and dicentrics was not significantly higher in the RS patients when compared to the NOR patients (Figure 5A). Based on the obtained results, we conclude that MN warrants evaluation as a potential marker of individual radiosensitivity in a larger cohort of RS patients. 

In line with the literature data [27,28], we found a significantly increased frequency of MN, dicentric and ring chromosomes after one week of RT (MN: *p* = 0.03; dicentrics: *p* = 0.0001; rings: *p* = 0.02). This increase also persisted one month after the end of RT (MN: *p* = 0.002; dicentrics: *p* = 0.00006; rings: *p* = 0.04) (Figure 5B). 

Next, we compared the level of CA/MN in cells of patients treated with CHT + RT and those treated solely with RT but did not find any significant differences between these groups of patients. However, we detected different kinetics in the accumulation of dicentrics (Appendix A). Namely, the amount of dicentrics increased significantly from a baseline of 1.9% to 4.57% after one week of RT in the RT alone patients (Fisher LSD, *p* = 0.001), while dicentrics increased from the baseline level of 1.83% to 2.82% in the PBL of BC patients treated with CHT + RT. Finally, at one month after the RT, we observed almost the same level of dicentrics in CHT + RT and RT alone patients, 4.3% and 4.5%, respectively, very similar to the results of Moquet et al., who observed about 5% of dicentrics in the BC patients before the last fraction of RT [28], suggesting the persistence of RT-induced dicentric chromosomes up to one month after the completion of RT. Similar to dicentrics, the accumulation of ring chromosomes during the first week of RT was also significantly higher compared to the baseline value in the group of RT alone treated patients (Fisher LSD, *p* = 0.04). This strong trend to a higher accumulation of CA in RT alone treated patients suggests that CHT treatment may induce a kind of adaptive response and, thus, decrease genomic instability induced by RT.

Finally, we performed correlation analysis of CA/MN and DNA repair foci in patients in which all these endpoints were analyzed, but no correlation between CA/MN and DNA repair foci was found. We also analyzed the potential difference in the level of CA/MN in 13 patients with (9 patients with hyperpigmentation)/without (4 patients) late healthy tissue side effects. Our results did not show any difference in the frequency of CA and CBMN in the PBL of patients with/without late tissue toxicity (Appendix A), suggesting a lack of correlation between CA and CBMN induced by RT and occurrence of late healthy tissue side effects. 

### 3.4. DNA Repair Foci in PBL from BC Patients after In Vitro Irradiation

In order to further evaluate the eventual difference between RS and NOR patients, we performed in vitro experiments, in which DNA repair foci were analyzed at 30 min, 2 h and 24 h after 2 Gy of γ-rays (Figure 6). While an increased level of endogenous foci was found before RT in the fresh PBL of RS patients, as described above, no such difference was detected in our in vitro experiments. The most probable explanation relies on the fact that the freezing/thawing procedure, which was applied in in vitro experiments, may eliminate the damaged cells with a higher number of DNA repair foci and, therefore, mask the difference between RS and NOR patients, which was established via in vivo examination. 

As expected, we observed a major increase in the γH2AX, 53BP1 and co-localized γH2AX/53BP1 foci levels 30 min after 2 Gy irradiation (Fisher LSD test, *p* = 0.00001, 0.00006, 0.00001, respectively) (Figure 7A–C). Most of the γH2AX, 53BP1 and co-localizing γH2AX /53BP1 foci vanished at 2 h (Fisher LSD *p* < 0.00001). We found a further decrease in the foci levels at 24 h as compared with 2 h (Fisher LSD *p* < 0.00001; for γH2AX, 53BP1 and co-localizing γH2AX /53BP1 foci). Even though almost all foci disappeared at 24 h after irradiation, their level was still significantly higher than the level of endogenous foci at the same time point (Fisher LSD, *p* = 0.0001, 0.005 and 0.0003 for γH2AX, 53BP1 and co-localizing foci, respectively) (Figure 6). Importantly, we found a significantly lower number of 53BP1 (Fisher LSD *p* = 0.001) and co-localizing γH2AX/53BP1 foci (Fisher LSD, *p* = 0.0001) in the PBL of NOR patients in comparison to the RS group 30 min after irradiation. As far as no difference was observed between γH2AX foci at this time point (Figure 7A–C) and because the accumulation of 53BP1 at the location of DSB follows H2AX phosphorylation, we hypothesize that the downstream part of the DNA damage response signaling pathway may be impaired in cells from RS patients. At 2 and 24 h post-irradiation, we did not find any difference between RS and NOR patients in either γH2AX or 53BP1 (Figure 6 and Figure 7). 

Thus, only early stages of DNA damage response may be impaired in cells from RS patients, while later stages seem to be unaffected. The ROC analysis of the DNA repair foci data obtained 30 min after irradiation revealed an AUC value of 1 for 53BP1 and co-localizing γH2AX/53BP1 foci (Appendix A). Calculation of the post hoc statistical power using the mean values and standard deviations of γH2AX/53BP1 foci per cell showed a very high statistical power of 1, suggesting that the sample size was adequate and results are statistically significant. We conclude that the primary 53BP1 foci induced within 30 min post-irradiation may be a potential biomarker for the assessment of individual RS via in vitro experiments prior to RT. However, these promising results warrant validation in a larger cohort of patients. Although primary foci defined under in vitro conditions may likely represent an efficient biomarker of RS, residual foci did not constitute a usable biomarker of RS in the in vitro conditions in contrast to the in vivo conditions. 

### 3.5. Apoptosis in PBL from BC Patients after In Vitro Ionizing Radiation Exposure

In the same set of in vitro experiments, as described above, we also quantified endogenous and radiation-induced apoptosis using the cytometry-based Annexin-V/7-AAD assay 0, 24 and 48 h after irradiation with 2 Gy. Our results revealed neither correlation between the residual 53BP1 foci and apoptosis nor a difference in the level of radiation-induced apoptosis between RS and NOR patients (Appendix A), in contrast to some reported data [14,29,30]. One important difference, which likely underlines an eventual inconsistence in the results, is the usage of frozen/thawed samples in our in vitro experiments. The freezing/thawing procedure induces apoptosis, and this induction can shadow the actual effect of RS on the apoptotic response. Although we observed a higher level of in vitro radiation-induced 53BP1 foci in RS patients, these differences did not lead to a modification of apoptotic response to irradiation. This result is in line with the literature data [29], suggesting that apoptosis correlates with residual but not primary 53BP1 foci. The abovementioned lack of difference in the apoptotic response between RS and NOR patients is understandable because we did not see a difference between RS and NOR patients in the level of residual foci. We conclude that the analysis of apoptosis in frozen/thawed cells, as defined by in vitro experiments prior to RT, is not suitable as a predictive biomarker of RS.

## 4. Discussion

In the present study, we combined the analysis of in vivo-defined DNA repair foci, apoptosis, CA and micronuclei. For the first time, we studied the possible effects of CHT on the DNA damage response induced by RT. Finally, we analyzed DNA repair foci and apoptosis induced by in vitro irradiation in the PBL of breast cancer patients in relation to individual radiosensitivity.

We studied γH2AX/53BP1 foci in 57 BC patients before, 24 h after the 4th fraction of RT, one month and one year after RT. In PBL taken before the RT, we found a higher level of γH2AX and significantly higher level of 53BP1 in RS patients, as compared to NOR patients. The elevated-level residual DNA repair foci suggest higher DNA damage in the RS patients. However, this increase in the level of DNA damage did not lead to apoptosis, probably due to the fact that in vivo apoptotic cells are phagocytized by macrophages and, thus, were not detectable in the bloodstream of patients. Other research groups have previously reported similar observations of increased levels of DNA damage in the PBL of patients, which develop severe acute normal tissue reactions following RT [3,31,32,33,34], although other studies did not find such an association [2,35,36]. However, neither previous studies nor our current research demonstrated sufficient sensitivity and specificity of the γH2AX/53BP1 foci analysis prior to RT for clinical identification of the RS patients. 

While a previous study by Markova et al. reported a negative correlation between the accumulation of late apoptotic/necrotic (LAN) and the level of 53BP1 and also positive correlation with individual RS [2], our current study did not reveal any correlation of DNA repair foci with the level of LAN cells. There are few methodological differences between studies, which may account for such inconsistency in the results. First, while patients were grouped into three groups, G0, G1 and G2-G3, by Markova et al. [2], we merged the G0 and G1 patients in order to gain a larger NOR group to compare with the group of RS patients, which consisted of G2 patients only. Second, while foci have previously been analyzed automatically using Metafer software with the custom-made classifier, we enumerated foci using a new semi-automated classifier, providing a possibility to manually correct the errors made by the automated classifier. In line with our current results, Djuzenova et al., reported a higher level of γH2AX/53BP1 foci in RS BC patients prior to RT [3]. One main discrepancy between ours and Djuzenova’s study is the significantly higher prevalence of the G2/G3 patients in Djuzenova et al.’s study. While we revealed relatively low acute toxicity, namely 5 G2 and 0 G3 RS patients out of 57 total patients, Djuzenova et al., reported 20/69 G2 and 6/69 G3 patients. The reason for such a huge difference in the prevalence of RS breast cancer patients, which was also reported in our previous study [2], remains unknown.

In order to increase the total number of RS patients in our study, we enrolled those treated with CHT before the start of RT. Although the occurrence of RS patients was very similar in those two groups (3 CHT + RT and 2 RT alone), our results revealed a higher initial level of γH2AX/53BP1 foci in the CHT + RT group that persisted up to 24 h after 4th fraction, suggesting higher genotoxicity of CHT as compared to RT. On the other hand, CHT induced some kind of adaptive response against radiation-induced effects that was observed in the accumulation of dicentric chromosomes during RT. In line with previous studies [2,3], we also detected a lower level of γH2AX and co-localizing γH2AX/53BP1 foci one year after the RT in comparison to the level before RT. We hypothesized that a higher level of genetic instability in patients diagnosed with cancer compared to healthy controls [3,34] could resume to the normal healthy state following successful cancer treatment. This process is observed as a lower level of γH2AX/53BP1 foci one year post-RT in comparison to data observed before the start of RT. 

To validate the results of our in vivo analysis, we performed a set of in vitro experiments with PBL from four RS and four matched NOR patients, in which we analyzed γH2AX/53BP1 foci 30 min, 2 h and 24 h after the 2 Gy of γ-rays in order to access primary and residual foci. Similar to the previously published data, the peak level of foci was detectable at 30 min post-IR [24,37]. For the detection of residual foci, the 24 h post-radiation timepoint was shown to be reliable, since all primary foci had already vanished and only residual foci remained until that time [37,38]. Our results revealed an increased level of primary 53BP1 and co-localizing γH2AX/53BP1 foci in PBL of RS patients, as analyzed 30 min after irradiation with 2 Gy. However, we acquired only 4 RS and 4 NOR patients in our in vitro experiments. Interestingly, the statistical power calculated post hoc based on the mean values and standard deviations of γH2AX/53BP1 per cell was 1, which is in line with the results of ROC analyses where the AUC was also 1, strongly suggesting statistical significance in our results. However, due to the limited number of enrolled RS patients, the obtained data should be further confirmed in a larger group of patients. On the other hand, the increased level of endogenous foci found in in vivo conditions was not confirmed through in vitro examination. The likely reason was the freezing/thawing procedure that preceded the in vitro experiments. This procedure could lead to the elimination of damaged cells that were not able to survive the stress caused by the freezing/thawing. Djuzenova et al. also showed a difference in the response of RS patients to in vitro radiation when they analyzed cells from breast cancer patients with an adverse acute skin reaction (G3) to RT [3]. Their results showed significantly increased radiation-induced γH2AX foci 30 min after 50 cGy and the protracted disappearance of γH2AX foci, as analyzed 24 h after 2 Gy compared to the group of BC patients with normal skin reaction (G0–1) [3]. Chua et al. observed a higher level of residual co-localizing γH2AX/53BP1 in RS patients 24 h after in vitro irradiation with 4 Gy of IR [31] but no difference in primary foci, which is in contrast to our data. Lobachevsky et al. backtracked G3-4 RS cancer patients and analyzed the levels of γH2AX foci at six different timepoints after in vitro irradiation of their PBL [6]. Subsequently, they plotted the unrepairable component of γH2AX foci against the foci repair rate and found both higher unrepairable component and decreased repair rate of γH2AX foci in the G3-G4 RS patients [6]. We conclude that in vivo-defined endogenous γH2AX/53BP1 foci before the start of RT and also in vitro radiation-induced primary foci analyzed in PBL could be suggested as biomarkers for assessing individual RS. We also analyzed the late healthy tissue side effects but did not observe any correlation with in vivo-defined foci level; however, the statistical power of this analysis was low. Our results support the reports by other research groups that also did not find any correlation between the level of DNA repair foci and late healthy tissue side effects [36,39,40]. However, some other studies reported differences between patients with late healthy tissue side effects and patients with normal response [32,41,42], suggesting that more data need to be accumulated to confirm the possibility of using γH2AX/53BP1 DNA repair foci for the prediction of late healthy tissue side effects induced by RT. 

We studied apoptosis before, during and after RT in the MNC populations, namely: lymphocytes, monocytes, neutrophils and CD34+ hematopoietic stem/progenitor cells. No difference was found between RS and NOR patients, which is in contrast to our previous study where we analyzed the percentage of late apoptotic/necrotic (LAN) lymphocytes using Trypan blue assay and observed correlation with individual RS [2]. We applied, here, the flow-cytometry-based assessment of LAN cells using 7AAD staining that provided the possibility to analyze many more apoptotic cells in comparison to the Trypan blue assay used in our previous study (ten thousand vs. hundreds of cells, respectively). In addition, in the current study, we analyzed LAN cells in whole blood in contrast to analysis of LAN cells after LSM separation of MNC, as carried out by Markova et al. [2]. Although no difference between RS and NOR patients was observed, we found a decreased percentage of live cells in neutrophils and HSPCs as compared to lymphocytes, suggesting that these populations are more sensitive to RT. However, the primary site for HSPCs is bone marrow, and these cells are released into the blood when needed. Thus, the number of apoptotic HSPCs in blood, defined by us, does not represent the total number of dead HSPCs in the body. Next, we did not observe any effect of CHT treatment on the level of apoptosis. This is due to the time difference between the CHT and RT that was usually at least one month. This time is likely sufficient for the removal of cells killed by CHT from the blood. Changes observed in the in vivo-defined CA/MN/foci were not accompanied by changes in apoptosis, which could be explained by the fact that all apoptotic cells underwent phagocytosis in the blood. Next, we performed an investigation of radiation-induced apoptosis using the Annexin/7-AAD assay through the set of in vitro experiments that did not show any difference between RS and NOR patient groups, as opposed to some literature data [14,30]. Inverse correlation between lymphocyte apoptosis after in vitro radiation exposure and chronic toxicity after RT have been observed in other studies with PBL from prostate cancer patients [43,44]. However, we did not observe any correlation between the chronic toxicity and in vitro radiation-induced apoptosis in our study, probably because the chronic toxicities after prostate RT are usually much more severe compared to breast RT. In conclusion, we found no evidence showing a difference in the level of in vivo- or in vitro-induced apoptosis between RS and NOR patients. 

For the first time, we assessed the effect of radiosensitivity and chemotherapy in the PBL of BC patients using CA and CBMN assays. Vinnikov et al. reviewed the available studies considering the possibility to predict individual radiosensitivity using CA and CBMN techniques and found both positive [21,31,45] and negative results [46,47] for both assays [48]. In our study, we observed a trend to a higher level of endogenous MN in RS patients prior the RT but no difference between the RS and NOR patients by monitoring ring and dicentric CA. It has previously been shown that chronic exposure to low doses of IR can elevate the levels of dicentric and ring CA, as shown in occupational workers [49]. Here, we also observed a significant increase in MN and dicentrics but not ring chromosomes in the PBL of breast cancer patients during the RT. In line with our data for dicentrics and MN, we also found an increased level of DNA repair foci after four fractions of RT. On the other hand, in contrast to radiation-induced DNA repair foci, which vanished after the end of RT, the elevated level of MN and dicentrics was still observed after the course of RT. Our results are in line with several studies considering other types of cancer [27,50,51,52]. Gamulin et al. studied dicentrics, rings and MN in cancer patients before, during and after the RT and observed the accumulation of dicentrics, rings and MN biomarkers induced by RT [50]. Moquet et al. analyzed the level of dicentrics in the PBL of cancer patients and saw an accumulation of dicentric chromosomes in all patients [28]. The BC patients accumulated the least number of dicentric chromosomes in comparison to other tumor locations, such as prostate, lung, endometrium, esophagus and colon, due to the lower volume of healthy tissue exposed to radiation. The frequency of dicentrics before the last fraction of RT observed by Moquet et al. correlates well with our data. Here, we detected the same frequency of dicentric chromosomes, around 4% in the PBL of BC patients one month after the RT, suggesting the persistence of RT-induced dicentric chromosomes up to one month after the RT. Hartel et al. analyzed dicentric chromosomes in the PBL of prostate cancer patients right after, one and 2.5 years after radiotherapy and found 6, 4 and 3% of dicentrics, respectively [53], which is very similar to our observations of 4% dicentrics one month after the RT in BC patients. The aforementioned observations suggested that analysis of dicentrics is the best biomarker to assess the damage induced by RT, since the elevated values of dicentrics and MN remained detectable up to 1 month after RT and up to 2.5 years after RT in other studies [50,51,53]. We also observed a higher level of dicentric chromosomes and MN in the cells of RS patients 24 h after the 4th fraction of RT that could be caused by the lower efficiency of DNA repair and subsequent higher damage affecting CA and MN. In conclusion, CA and CBMN assays may have great value for studying the genomic instability induced by ionizing radiation since it accumulates and preserves for a longer time compared to DNA repair foci. A consideration of the mixed results in the available studies’ application for the assessment of individual radiosensitivity must be confirmed in a larger study with a higher number of patients.

## 5. Conclusions

In this study, we found a significantly higher amount of endogenous γH2AX and 53BP1 DNA repair foci in fresh PBL withdrawn prior to RT from the G2 RS patients in comparison to the G0 - 1 NOR patients, while this effect was not reproduced by analyzing frozen-thawed cells. A higher level of in vitro radiation-induced primary 53BP1 DNA repair foci was also found in the PBL of RS patients as compared to the NOR patients. Based on our data and the literature data, we conclude that a number of experimental parameters should be considered while DNA repair foci are being potentially used for the assessment of individual RS. Analysis of apoptosis, either in vivo or in vitro, was not shown to be a predictive biomarker for assessing individual RS in our study. Only a trend to higher baseline levels of micronuclei in RS patients was revealed, while this increase was not significant. MN and CA analysis was able to detect the accumulation of damage during RT and may find application in assessing the risk of long-term IR-induced DNA damage. A combination of several molecular biomarkers may provide a sensitive tool for assessing the individual radiosensitivity.

## Figures and Tables

**Figure 1 biomedicines-11-01122-f001:**
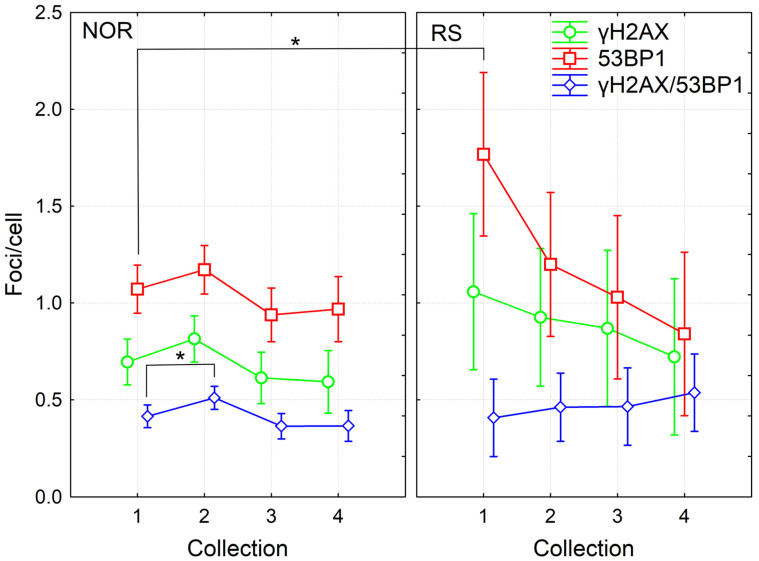
DNA repair γH2AX /53BP1 foci in PBL from BC patients collected before (1), 24 h after 4th fraction (2) and after RT (one month (3), one year (4)). The mean numbers from 52 NOR (G0-G1 acute skin toxicity) and 5 RS (G2 acute skin toxicity) patients along with 95% confidence interval are shown. *p*-values are indicated by asterisk (* *p* ≤ 0.05).

**Figure 2 biomedicines-11-01122-f002:**
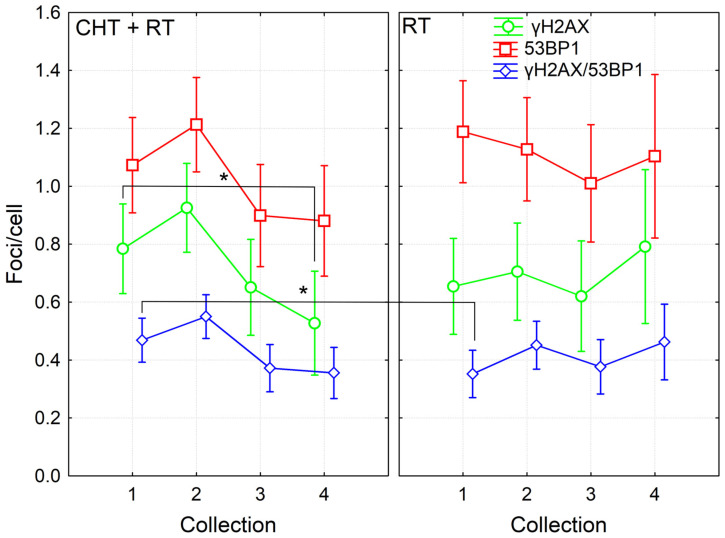
DNA repair γH2AX /53BP1 foci in PBL from BC patients collected before (1), 24 h after 4th fraction (2) and after RT (one month (3), one year (4)) from patients treated with either RT (twenty-six patients) or CHT+RT (thirty-one patients). Data are shown as mean with 95% confidence interval. *p*-values are indicated by asterisk (* *p* ≤ 0.05).

**Figure 3 biomedicines-11-01122-f003:**
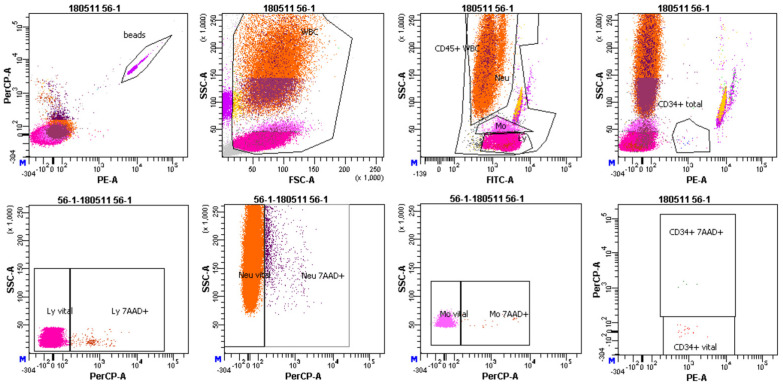
Representative image of FACS gating. First, white blood cells (WBCs) were gated. Afterwards, lymphocytes (Ly), monocytes (Mo) and neutrophils (Neu) were gated according to CD45 expression as observed in the FITC-A fluorescent channel. In the next step, CD34+ hematopoietic stem/progenitor cells were gated according to the expression of CD34 antibody as observed in the PE channel. Lastly, percentage of late/apoptotic cells (7AAD+), as shown in PerCP—a fluorescent channel was assessed in all cell populations.

**Figure 4 biomedicines-11-01122-f004:**
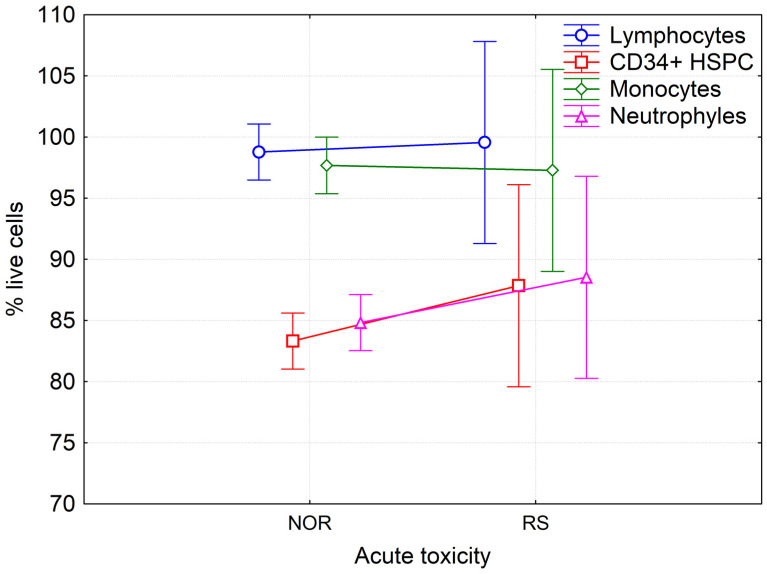
Percentage of live lymphocytes, monocytes, neutrophils and hematopoietic stem/progenitor cells in peripheral blood of BC patients before RT. Mean values from forty-seven NOR patients (G0-G1) and four RS (G2) patients along with 95% confidence interval are shown.

**Figure 5 biomedicines-11-01122-f005:**
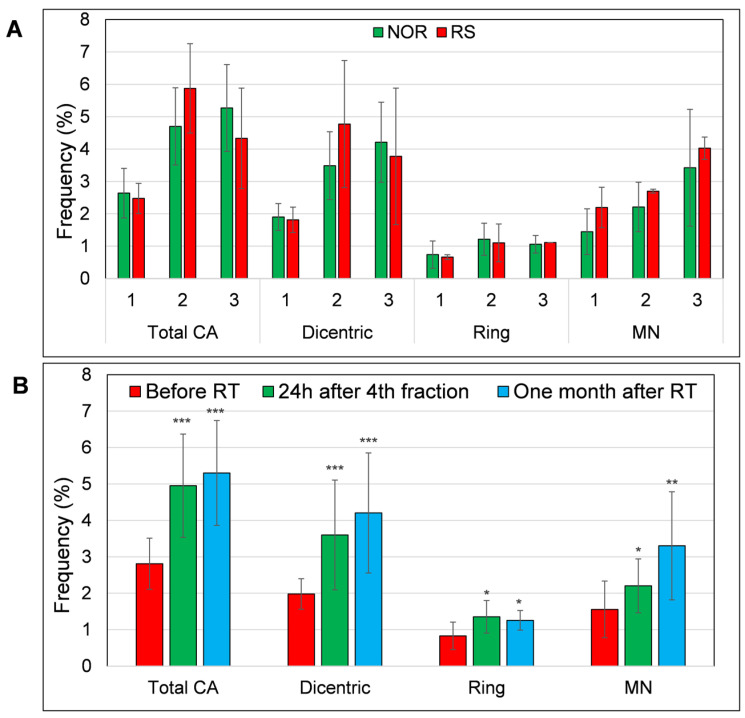
The level of chromosomal aberrations (dicentric, ring chromosomes and total aberrations (summation of dicentrics and rings)) and MN (**A**) in the PBL of RS and NOR patients and (**B**) level of CA and MN in all samples collected before (1), 24h after 4th fraction (2) and one month after RT (3). Frequency of CA and MN was defined as percentage of aberrant cells per 100 metaphases/MN per 100 binucleated cells (1000 metaphases/binucleated cells were analyzed, respectively). Data from 11 NOR and 2 RS patients are shown as mean and standard deviation. *p*-values are indicated by asterisk (* *p* ≤ 0.05; ** *p* ≤ 0.01; *** *p* ≤ 0.001).

**Figure 6 biomedicines-11-01122-f006:**
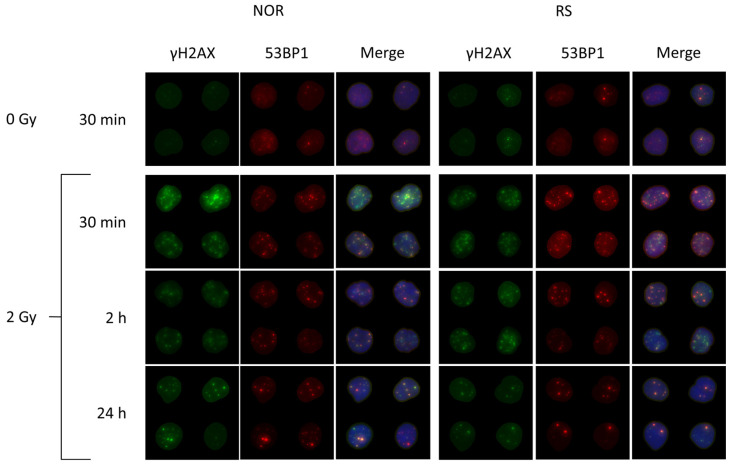
Representative images of radiation-induced DNA repair foci in the PBL from BC patients. The cells were collected before the start of RT and frozen in liquid nitrogen. After thawing and preincubation, the cells were irradiated in vitro with 0 and 2 Gy of γ-rays and then analyzed 30 min, 2 h and 24 h post-irradiation. γH2AX in green, 53BP1 in red and merged images of co-localized γH2AX/53BP1 foci with DNA stained with DAPI in blue color are shown.

**Figure 7 biomedicines-11-01122-f007:**
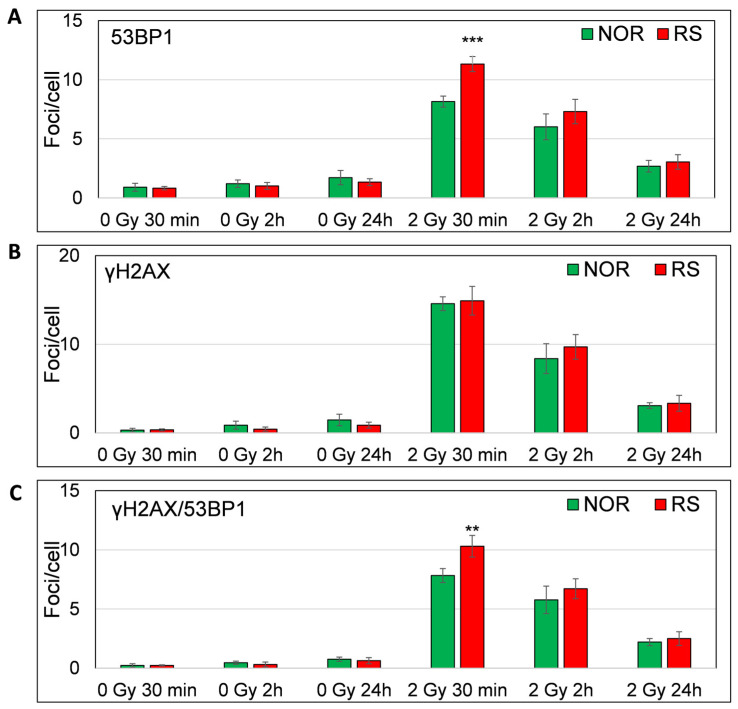
In vitro assessment of γH2AX /53BP1 DNA repair foci. Figure shows the level of 53BP1 (**A**), γH2AX (**B**) and co-localized γH2AX/53BP1 (**C**) foci 30 min, 2 h and 24 h after irradiation with 0 and 2 Gy of γ-rays. Mean values from four RS and four matched NOR patients (No. 24, 26, 30 and 40 from Appendix A) are shown. *p*-values are indicated by asterisk (** *p* ≤ 0.01; *** *p* ≤ 0.001).

## Data Availability

All data are available upon request.

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
