# Peer review of "Assessment of Individual Radiosensitivity in Breast Cancer Patients Using a Combination of Biomolecular Markers"

_biomedicines, 2023, doi:10.3390/biomedicines11041122_

Round 1

Reviewer 1 Report

Authors:  Matus Durdik, Eva Markova, Pavol Kosik, et al.  

Title:  Assessment of individual radiosensitivity in breast cancer patients using combination of biomolecular markers 

COMMENTS: 

The Authors compared certain events/consequences in the radiation response of lymphocytes from peripheral blood of breast cancer patients with normal or elevated radiosensitivity. It seems that the Authors aimed at revealing some specific differences that would be used as predictive markers of the higher radiosensitivity in normal tissues of patients with breast cancer. Basing on their results obtained, the Authors only sugest that γH2AX/53BP1 foci and micronuclei (MN) may be considered as potential biomarkers of individual radiosensitivity. 

There are some points of criticism: 

After reading the manuscript, it seems unclear in what the value of this study is (?). Similar studies with similar goals have been performed before and their results have been published (see: DOI: 10.1515/znc-2005-11-1219  ; doi: 10.1080/09553002.2016.1230238.   [ref 46 in the manuscript]; doi: 10.1667/RR0746.1. ; doi: 10.1186/1748-717X-5-85. ;  doi: 10.1016/j.ijrobp.2013.11.218. [ref 32 in the manuscript]). The Authors should explain in which points their own study is more advanced and show new findings as compared with those publications. 

The Authors describe some phenomena they discovered but without deep exploring it. It is unclear how the revealed higher levels of γH2AX/53BP1 foci may explain the elevated radioresistance (or the suggested impairment in DNA damage response), while no differences were found in the levels of residual foci and appoptosis. By the way, it would be nice to compare the γH2AX/53BP1 foci data with the data of comet assay. 

 In Introduction, the Authors should describe in more detail the risk of post-radiation complications in normal tissues of patients with breast cancer (in particular, the risk of pulmonary fibrosis). 

Author Response

Reviewer 1 

The Authors compared certain events/consequences in the radiation response of lymphocytes from peripheral blood of breast cancer patients with normal or elevated radiosensitivity. It seems that the Authors aimed at revealing some specific differences that would be used as predictive markers of the higher radiosensitivity in normal tissues of patients with breast cancer. Basing on their results obtained, the Authors only suggest that γH2AX/53BP1 foci and micronuclei (MN) may be considered as potential biomarkers of individual radiosensitivity. 

There are some points of criticism: 

After reading the manuscript, it seems unclear in what the value of this study is (?). Similar studies with similar goals have been performed before and their results have been published (see: DOI: 10.1515/znc-2005-11-1219  ; doi: 10.1080/09553002.2016.1230238.   [ref 46 in the manuscript]; doi: 10.1667/RR0746.1. ; doi: 10.1186/1748-717X-5-85. ;  doi: 10.1016/j.ijrobp.2013.11.218. [ref 32 in the manuscript]). The Authors should explain in which points their own study is more advanced and show new findings as compared with those publications. 

R: We agree that similar studies have previously been performed, but there is difference which makes our study unique. Specifically, the fact that it presented the combination of various biomarkers such as in vivo-defined foci, apoptosis, chromosomal aberrations and micronuclei as well as in vitro radiation induced foci and apoptosis. It is also beneficial to perform similar studies in different countries, due to differences in radiation protocol, side effects assessment and haplotype present in selected population. Innovation in our study is also in analyzing the effect of CHT along with RT on the level of genomic instability in PBL of BC patients.

Advancements of our study are now explained in the Introduction and also in the beginning of the discussion part.

The Authors describe some phenomena they discovered but without deep exploring it. It is unclear how the revealed higher levels of γH2AX/53BP1 foci may explain the elevated radioresistance (or the suggested impairment in DNA damage response), while no differences were found in the levels of residual foci and apoptosis. By the way, it would be nice to compare the γH2AX/53BP1 foci data with the data of comet assay. 

R: In our study we observed higher level of endogenous DNA repair foci in radiosensitive (RS) patients exhibit higher level of residual DNA repair foci that suggest higher DNA damage in the RS patients. However, this increase in the level of DNA damage did not lead into apoptosis, probably due to fact that in vivo apoptotic cells are phagocytized by macrophages thus was not detectable in the bloodstream of patients.

According to the in vitro experiments, radiosensitive patients showed higher level of primary foci as observed 30 minutes after irradiation, but no difference in the residual foci as observed 24h after IR, suggesting higher level of DNA damage in RS patients however DNA repair is not impaired in these patients. Impairment in DNA repair would the most probably also lead to differences in the apoptotic response that was not observed here. Our data suggest increased DNA damage and impaired DNA repair signaling process with overactivation of 53BP1 at the sites of damage in PBL of RS patients

Comet assay after in vitro irradiation was performed in previously published papers with head and neck cancer patients (Rzeszowska-Wolny et al., 2008), where they found weak, statistically significant correlation between in vitro radiosensitivity measured as the rate of DNA damage repair using comet assay and the cumulative radiation dose exerting the maximum acute reaction using Dische scale and also breast cancer patients (Sterpone et al., 2010) reduced repair ability was found in BC patients showing high degrees of tissue side effects as classified using RTOG/EORC criteria. Our results correlate with these data because we have also observed higher level of in vitro radiation – induced DNA damage in the patients exhibiting healthy tissue side effects using DNA repair foci.

Considering in vivo defined DNA damage there were only few studies analyzing comet assay to predict individual radiosensitivity based on residual DNA damage and they were no table to find out the difference between RS and NOR patients (Gamulin et al., 2010; Greve et al., 2012), therefore we have chosen to use DNA repair foci instead of comet assay for our study.

Statement discussing radiosensitivity and apoptosis was added into discussion part.

 In Introduction, the Authors should describe in more detail the risk of post-radiation complications in normal tissues of patients with breast cancer (in particular, the risk of pulmonary fibrosis). 

R: We have described the risk of pulmonary fibrosis in the Introduction, based on reviewer’s suggestions.

Reviewer 2 Report

General Comments

The manuscript entitled “Assessment of individual radiosensitivity in breast cancer patients using combination of biomolecular markers” aims to establish biomarkers to identify radiotherapy patients with increased radiosensitivity. The authors tested a range of endpoints assayed in cells from blood samples obtained from radiotherapy patients and attempted to correlate the output with radiosensitivity status of the patients. The investigation presents a genuine interest for a broad audience of readers and may have a potential impact, along with other studies in this field, on clinical radiotherapy practice.

The most controversial observation reported in the manuscript is that in radiosensitive (RS) patients group an increased level of endogenous DNA 53BP1 foci was observed before the start of radiotherapy (RT), as compared to normal responding patients (NOR) (Figure 1). Associated with this observation is a conclusion that this biomarker (lines 468-469) could be suggested for assessing individual RS. This observation is even more surprising considering that in blood samples obtained one year after RT the level of these foci is much lower. Unfortunately, no statistical significance for this difference (one year post-RT vs pre-RT) is provided.  The interpretation of this trend should be addressed in the discussion of the manuscript: is it just a statistical variation due to low number of patients in the group or there is some sort of prolonged adaptive reaction in response to RT?

Specific comments

1.       In methodological context, the reasons for choosing of 24 hr interval after RT fraction for the second point (Figure 1) are not clear and no justification is given for this choice. Since at this time interval, the majority of DNA repair foci induced by RT will be eliminated, and therefore with shorter interval more difference could be observed between pre- and post- RT points for co-localised foci, which could possible results in statistical significance of difference between these points for separate CHT+RT and RT analysis. Addressing this point would be beneficial for the manuscript.

2.       Section 2.3 in Material and Methods should be split into two sections – one describing patients and another one describing cell processing. Numbers of RS and NOR should be stated in patients’ section. Table 1 should go to supplementary materials since majority of particular details of individual patients are not used in the main body of the manuscript.

3.       The legend for Figure 3 lacks details. Interpretation of diagrams is difficult for the reader unless clear association between axes names and fluorescent labels used to discriminate cell populations is specified.

4.       There is an apparent inconsistency between statement in line 270 – “lower level of LAN cells in neutrophils … and HSPC” with results presented in Figure 4. Should it be “lower level of viable cells” ? Figure 4 shows data for “before RT” group, while statement in lines 263-265 refers to all for time points and Figure 4. Clarification is required.

5.       As stated in line 375, ROC analysis of 53bp1 and co-localising foci 30 min post in vitro blood irradiation revealed AUC = 1. In general ROC analysis, AUC = 1 means achieving an ideal classifier. Since it cannot be claimed that the reported here biomarkers are ideal classifiers due to low number of studied subjects (4 + 4) for this comparison, these numbers should be indicated in the context of AUC in addition to  the statement that large cohort studies are required.

 Minor comments

Line 101 – the sentence is not complete.

Author Response

Reviewer 2

General Comments

The manuscript entitled “Assessment of individual radiosensitivity in breast cancer patients using combination of biomolecular markers” aims to establish biomarkers to identify radiotherapy patients with increased radiosensitivity. The authors tested a range of endpoints assayed in cells from blood samples obtained from radiotherapy patients and attempted to correlate the output with radiosensitivity status of the patients. The investigation presents a genuine interest for a broad audience of readers and may have a potential impact, along with other studies in this field, on clinical radiotherapy practice.

The most controversial observation reported in the manuscript is that in radiosensitive (RS) patients group an increased level of endogenous DNA 53BP1 foci was observed before the start of radiotherapy (RT), as compared to normal responding patients (NOR) (Figure 1). Associated with this observation is a conclusion that this biomarker (lines 468-469) could be suggested for assessing individual RS. This observation is even more surprising considering that in blood samples obtained one year after RT the level of these foci is much lower. Unfortunately, no statistical significance for this difference (one-year post-RT vs pre-RT) is provided.  The interpretation of this trend should be addressed in the discussion of the manuscript: is it just a statistical variation due to low number of patients in the group or there is some sort of prolonged adaptive reaction in response to RT?

R: While there is a limited number of RS patients, we observed a statistically significant decrease of gH2AX foci one-year post RT vs. pre-RT in patients treated with CHT (line 231-234) and a strong trend to decreased level of co-localizing gH2AX/53BP1 foci one-year post RT vs. pre-RT was present in the group of all patients (Fisher LSD, p = 0,059).  

We hypothesized that higher level of genetic instability in patients diagnosed with cancer compared to healthy controls [3, 34] could resume to the normal healthy state when the cancer treatment is done. This process is observed as a lower level of γH2AX/53BP1 foci one-year post RT in comparison to data observed before the start of RT.

Correspondent change was done in the discussion part.

Specific comments

  1. In methodological context, the reasons for choosing of 24 hr interval after RT fraction for the second point (Figure 1) are not clear and no justification is given for this choice. Since at this time interval, the majority of DNA repair foci induced by RT will be eliminated, and therefore with shorter interval more difference could be observed between pre- and post- RT points for co-localized foci, which could possible results in statistical significance of difference between these points for separate CHT+RT and RT analysis. Addressing this point would be beneficial for the manuscript.

R:  For several reasons it was not possible to took patients’ blood directly after irradiation. The dose acquired by blood cells during irradiation is not homogenous because of the blood circulation and the time of reaching equilibrium of irradiated and non - irradiated cells is also individual. The maximum amount of foci is observed up to 30 minutes after irradiation, but the majority of foci are eliminated even 2 hours after irradiation. However, it is known that the fraction of the foci remained on the damage site 24hours after irradiation, so called residual foci, and we aimed to target these types of foci. Therefore, we choose to draw the blood next morning when relative equilibrium in blood and plateau in kinetics of residual foci is usually observed.

Appropriate changes were done in the material and method section.

  1. Section 2.3 in Material and Methods should be split into two sections – one describing patients and another one describing cell processing. Numbers of RS and NOR should be stated in patients’ section. Table 1 should go to supplementary materials since majority of particular details of individual patients are not used in the main body of the manuscript.

R: The sections were split accordingly and the Table was moved into the supplement.

  1. The legend for Figure 3 lacks details. Interpretation of diagrams is difficult for the reader unless clear association between axes names and fluorescent labels used to discriminate cell populations is specified.

R: Detailed description of used fluorophores in different populations was added to the Figure 3 legend.

  1. There is an apparent inconsistency between statement in line 270 – “lower level of LAN cells in neutrophils … and HSPC” with results presented in Figure 4. Should it be “lower level of viable cells” ? Figure 4 shows data for “before RT” group, while statement in lines 263-265 refers to all for time points and Figure 4. Clarification is required.

R: The manuscript has been revised according to reviewer’s comments. Figure 4 shows only one time-point because there was no difference between RS and NOR patient in any of the time-points. Also, no difference between time-points was found. So, we showed only the data from the time-point in which the difference in the foci level was found.

  1. As stated in line 375, ROC analysis of 53bp1 and co-localising foci 30 min post in vitro blood irradiation revealed AUC = 1. In general ROC analysis, AUC = 1 means achieving an ideal classifier. Since it cannot be claimed that the reported here biomarkers are ideal classifiers due to low number of studied subjects (4 + 4) for this comparison, these numbers should be indicated in the context of AUC in addition to the statement that large cohort studies are required.

R: We completely agree that the reported biomarkers are not ideal classifiers for assessing individual radiosensitivity. However, we were not able to obtain high number of RS patients. When we calculated the post-hoc statistical power of our results based on the mean value of gH2AX/53BP1 foci per cells and standard deviations we get statistical power = 1 as calculated through online calculator www.clincalc.com, suggesting statistical significance of ours results. But we agree that large cohort studies are absolutely needed.

Appropriate addition was made to the Results and Discussion part of the revised manuscript.

 Minor comments

Line 101 – the sentence is not complete.

R: It was a typo, we finished the sentence now. “Five G2 patients exhibited severe acute side effects (erythema, skin desquamations) as assessed by the physicians in the NCI.”

Round 2

Reviewer 1 Report

Thanks for responding to my criticism. Good luck in your research!